# CIRP: Cross-Item Relational Pre-training for Multimodal Product Bundling

## ABSTRACT

Product bundling has been a prevailing marketing strategy that is beneficial in the online shopping scenario. Effective product bundling methods depend on high-quality item representations capturing both the individual items' semantics and cross-item relations. However, previous item representation learning methods, either feature fusion or graph learning, suffer from inadequate cross-modal alignment and struggle to capture the cross-item relations for cold-start items. Multimodal pre-train models could be the potential solutions given their promising performance on various multimodal downstream tasks. However, the cross-item relations have been under-explored in the current multimodal pre-train models.

To bridge this gap, we propose a novel and simple framework Cross-Item Relational Pre-training (CIRP) for item representation learning in product bundling. Specifically, we employ a multimodal encoder to generate image and text representations. Then we leverage both the cross-item contrastive loss (CIC) and individual item's image-text contrastive loss (ITC) as the pre-train objectives. Our method seeks to integrate cross-item relation modeling capability into the multimodal encoder. Therefore, even for cold-start items without explicit relations, their representations are still relation-aware. Furthermore, to eliminate the potential noise and reduce the computational cost, we harness a relation pruning module to remove the noisy and redundant relations. We apply the item representations extracted by CIRP to the product bundling model ItemKNN, and experiments on three e-commerce datasets demonstrate that CIRP outperforms various leading representation learning methods. Our code and dataset will be released upon acceptance.

## CCS CONCEPTS

• **Information systems → Multimedia and multimodal retrieval**; **Recommender systems**.

## KEYWORDS

Multimodal Bundle Construction, Bundle Recommendation, Multimodal Pre-train, Vision Language Model

## 1 INTRODUCTION

Product bundling is a prevailing strategy in retail markets, which aims to promote sales and improve consumer satisfaction by combining a set of products into bundles. Especially in the era of online

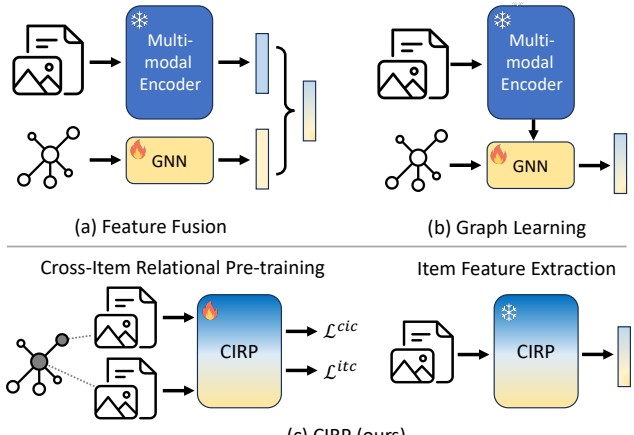

**Figure 1: Comparison of three item representation learning paradigms of incorporating semantic and relational data. Different from Feature Fusion and Graph Learning, our method features at integrating the relational info into the multimodal encoder.**

shopping, an extensive array of products from diverse retail sectors are gathered on a single platform, thereby boosting bundling opportunities. Albeit beneficial, the ever-growing number of products poses challenges to the task of product bundling, which has also attracted particular attention in the research community.

In order to develop an effective product bundling model, the challenge is to learn high-quality representations of product items from two aspects. (1) The representations need to capture the rich semantic features of individual items. For instance, the textual description of an electronic device represents its brand and functions, while the image of fashion apparel highlights intricate details, such as print and texture. Such multimodal semantics are essential for bundling items that are similar or compatible in terms of functionality or aesthetics. (2) The representations should be able to model the diverse and implicit relationships among items. Such relations (*e.g.,*, co-purchasing or sequential interaction) imply auxiliary but crucial information to bundle certain items. For example for the famous *beer and diaper* case [1], they can hardly be put together if only semantic features are considered, while the co-purchase relationship implies that they could be a good bundling option. Given the pervasive success of representation learning models pre-trained on large-scale dataset [5, 12, 33], it is promising to pre-train an item representation learning model based on large-scale e-commerce data.

Considering the two aspects of semantic and relational modeling, current pre-training methods can broadly be categorized into two

---

[1]https://en.wikipedia.org/wiki/Association_rule_learning

streams: feature fusion and graph learning, as shown in Figure 1. Specifically, feature fusion-based methods [11, 38] learn item representations of each modality separately (*e.g.,* , visual, textual, and relational) and subsequently fuse them into a multimodal representation. Graph learning-based methods [19, 37] use pre-extracted item semantic features as node attributes and apply graph learning algorithms to refine these features through graph propagation. In spite of their wide usage, both of these methods suffer from two main limitations. First, the features from multiple modalities are learned separately, lacking in-depth cross-modal alignment or enhancement. Second, they struggle with accurately capturing relational patterns for cold-start items, which are new items that have not yet established any relational data with other items. In the light of these limitations, the powerful multimodal foundation models [22, 33] could be the potential solution given their outstanding performance on various multimodal downstream tasks. However, they focus mostly on the semantic modeling of image and text, with limited exploration into cross-item relations.

To bridge this gap, we propose a novel but simple framework, named Cross-Item Relational Pre-training (CIRP). Specifically, we first construct an item-item relational graph based on the co-purchase item pairs. From this graph, we sample a pair of items that are directly connected and adopt two unimodal encoders, *i.e.,* , image and text encoders [22], to generate each item's multimodal representations of image and text. Thereafter, we harness the cross-item contrastive (CIC) loss to enforce the representations of related items to be close to each other. Simultaneously, we keep the image-text contrastive (ITC) loss of individual items to retain the cross-modal alignment. This simple pre-training framework can naturally integrate cross-item relations into the multimodal encoder (by the CIC loss), while preserving the in-depth aligned multimodal semantics (by the ITC loss). More importantly, even for cold-start items, CIRP can generate relation-aware multimodal representations, which could hardly be achieved by previous item representation learning methods of feature fusion and graph learning. Furthermore, considering the potential noise and heavy computational cost brought by the extensive amount of item-item relations, we propose a novel relation pruning module to remove those noisy or redundant connections. In the downstream task of product bundling, we use CIRP to extract item representations, which are then fed to the product bundling model ItemKNN. Experiments on three large-scale e-commerce show that CIRP can significantly boost the performance of product bundling compared with various leading methods for item representation learning. More interestingly, when pruning 90% of the relations, our method only experiences a slight performance drop, while just taking 1/10 of the pre-training time. The main contributions of this work are summarized as follows:

- To the best of our knowledge, we are among the first to integrate the cross-item relational information into a multimodal pre-train model for product bundling.
- We develop a novel framework CIRP that can simultaneously model both individual item's semantics and cross-item relations. We also propose a relation pruning module to improve pre-training efficiency and efficacy.

- Experimental results on three product bundling datasets demonstrate the competitive performance of our method in terms of both efficacy and efficiency.

## 2 RELATED WORK

We briefly review the works related to this paper from two streams: 1) multimodal pre-training, and 2) product bundling.

### 2.1 Multimodal Pre-training

Multimodal pre-training has experienced enormous progress in recent years. CLIP [33] is the pioneering work that unleashes the power of large-scale multimodal data by using a simple cross-modal contrastive loss. Following this trend, a surge of multimodal pre-training works emerge, such as BLIP [22] and BEiT-3 [42], which achieve impressive performance on various downstream tasks. Especially after the breakthrough of Large Language Models (LLMs) brought by ChatGPT, researchers swiftly grasp this opportunity by integrating the language understanding capability of LMMs into the multimodal models, such as BLIP-2 [21], LLaVA [26, 27], MiniGPT-4 [49], *etc.* These multimodal pre-trained models have demonstrated impressive performance on various downstream vision-language tasks. However, these models are not specifically designed to capture the relational data.

Multiple studies in the E-commerce domain consider both multimodal semantic and relational pre-training, including K3M [50], KnowledgeCLIP [32], KG-FLIP [17], FashionKLIP [44], *etc.* The main objective of these works is to incorporate knowledge graph into the pre-trained multimodal models. Nevertheless, the relations used in these works are the triplet-formatted facts, while the cross-item relations are not captured. Consequently, these models can only be used for multimodal understanding-based tasks, such as classification, caption, cross-modal retrieval, *etc.*, while the tasks (*e.g.,* product bundling) that require the modeling of cross-item relations cannot be tackled well.

### 2.2 Product Bundling

Product bundling aims to combine a list of individual products into a bundle. It has been widely used in various business sections, such as e-commerce [39, 40], fashion [30], music stream [31], games [4], trips [25], and food [24], *etc.* Conventional product bundles are manifested by retailers manually, which is applicable due to a relatively small number of items. With the explosion of E-commerce, the number of items on a single platform boosts drastically, necessitating automatic product bundling methods. There are multiple works that are developed for personalized product bundling [1–3, 7, 14, 45]. However, most of them only rely on the relations between items, thus they cannot deal with cold-start items. Therefore, it is necessary to incorporate the multimodal semantic features of items and build multimodal product bundling methods [31]. We deem that the key of multimodal product bundling is to learn high-quality relation-aware multimodal representation, which is reasonably to be achieved by a pre-train model. To evaluate the performance of pre-train model, we leverage a simple method, ItemKNN [36], which is also adopted by the previous work [40] for product bundling.

Existing pre-train methods that can capture both cross-item relational and semantic data are the graph learning models, including

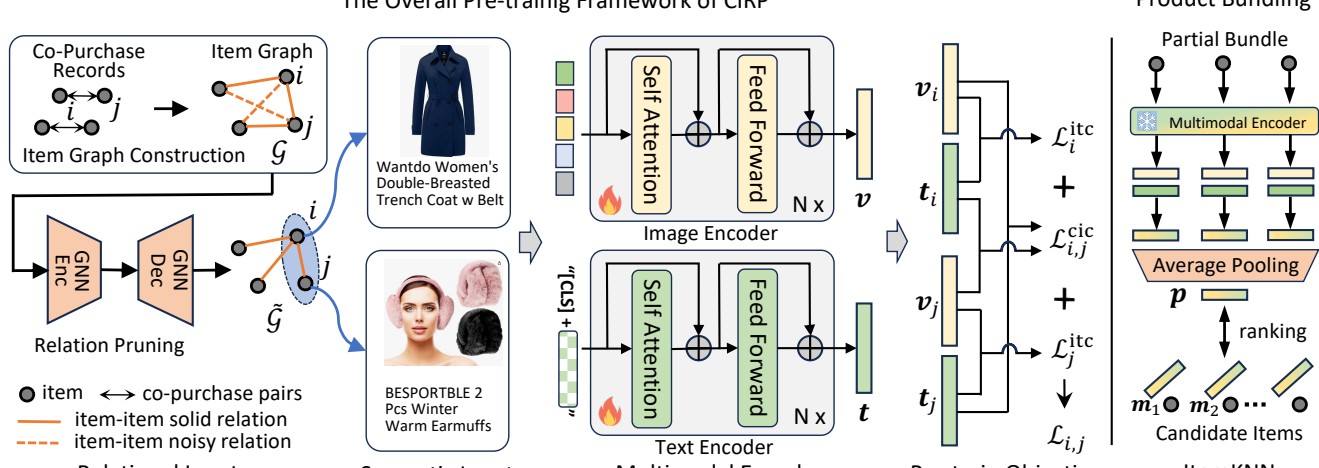

**Figure 2: Illustration of the overall pre-training framework (CIRP) and the downstream task of product bundling. CIRP takes relational and multimodal semantic inputs, leverages a multimodal encoder, and is optimized by the CIC and ITC losses. For the downstream task, we leverage the ItemKNN model and CIRP extracted item representations for product bundling.**

GCN [20], GPT-GNN [16], HeCo [43], MPKG [10], *etc.* These methods use pre-extracted item features to initialize the node representation, and then leverage graph propagation to integrate the relational information. They focus more on the relational information, falling short in modeling the semantics, especially the cross-modal alignment. Even worse, for cold-start items that have no relations before, such graph learning methods can hardly endow any relational data into the item representation. Our work would attempt to integrate the relational information into the multimodal backbone model with the relational pre-training framework.

To be noted, our task targets at bundle construction rather than personalized bundle recommendation [9, 29, 30, 34, 46, 48], which aims to recommend pre-defined bundles to users according to users' preference. Nevertheless, these two tasks are highly related to each other since bundle construction is a pre-requisite step for platforms before they can offer bundle services.

## 3 METHOD

We first introduce the preliminary of the pre-training and the downstream task. Then, we present our proposed CIRP, as shown in Figure 2, which includes the relation pruning module and the pre-training framework. Finally, we briefly describe how to use the pre-trained representations for the task of product bundling.

### 3.1 Preliminary

**Problem Formulation:** We target at pre-training a multimodal item representation learning model for the downstream task of product bundling. Given a large set of product items $\mathcal{I} = \{i_n\}_{n=1}^{N}$, associated with each item's semantic features (*i.e.,* text $\mathcal{T} = \{t_i\}$ and image $\mathcal{V} = \{v_i\}$, where $i \in \mathcal{I}$) and item-item relations (*i.e.,* an item-item graph, will be introduced later), we aim to train a model $\Phi(v, t; \Theta)$ that consumes an item's image $v$ and text $t$, then outputs a multimodal representation $(\mathbf{v}, \mathbf{t})$ that endow both semantic and

relational information, where $\Theta$ is the parameters of the representation learning model. Thereafter, the item representation can be incorporated into product bundling models.

**Item Graph Construction:** There are multiple types of relations that could be considered for pre-training, such as co-purchase (two items that are frequently purchased together), sequential interaction (two items are interacted with a certain user consecutively), and knowledge graph (two items are related via the knowledge graph), *etc.* Following the work [40], we construct the item-item relation graph based on the co-purchase data, which has been demonstrated to be helpful relations for product bundling. Specifically, if two items are purchased consecutively by the same user in a short time period (*i.e.,* one day), these two items will be connected with an undirected edge. Thereby, all the items $\mathcal{I}$ and edges $\mathcal{E}$ form a homogeneous graph $\mathcal{G} = \{\mathcal{I}, \mathcal{E}\}$. It is worth noting that we aim to develop a pre-training framework for general types of relations among items. Different relation types or graph construction details may impact the pre-training, which will be left for future work.

### 3.2 Relation Pruning

The graph is prone to include noisy or redundant edges, which are useless or even harmful to cross-item relational modeling. It is imperative to remove such relations from the item-item graph. More importantly, given the extensive amount of item-item relations, it is computationally heavy for large-scale pre-training. Therefore, to enhance the quality of the item-item relation graph and accelerate pre-training, we propose a relation pruning module. It comprises of two steps: we first train a graph auto-encoder to obtain the node representations; then we use the learned node embedding to prune the original graph.

*3.2.1 Graph Auto-Encoder.* The graph auto-encoder is devoted to learning informative node embeddings that well preserve the graph structural information. Thereby, we follow the previous

works [6, 13] and use the LightGCN kernel as the graph encoder. It includes two main components: information propagation and layer aggregation. The information propagation is defined as:

$$\mathbf{e}_i^{(k)} = \frac{1}{|\mathcal{N}_i|} \sum_{j \in \mathcal{N}_i} \mathbf{e}_i^{(k-1)}, \tag{1}$$

where $\mathcal{N}_i$ denotes the neighbors of item $i$ in the graph, $\mathbf{e}_i^{(k)} \in \mathbb{R}^d$ is the $k$-th layer representation of item $i$ during information propagation, and $d$ is the dimensionality of the representation. The first layer of the node embedding $(\mathbf{e}_i^{(0)})$ is randomly initialized. After $K$ layers of information propagation, we aggregate the representations of multiple layers and obtain the final item representation $\mathbf{e}_i$, represented as:

$$\mathbf{e}_i = \sum_{k=0}^{K} \mathbf{e}_i^{(k)}. \tag{2}$$

The decoder part is a typical link prediction task. Given a pair of items $(i, j)$, we use the inner product to calculate the score of how likely these two items could be connected to each other, formally represented as: $s_{i,j} = \mathbf{e}_i^{\top} \mathbf{e}_j$. We resort to the Bayesian Personalized Ranking (BPR) [35] loss to optimize the model, and the loss used to train the graph auto-encoder is denoted as:

$$\mathcal{L}_G = \sum_{(i,j) \in \mathcal{E}, (i,j') \notin \mathcal{E}} -\ln\sigma(s_{i,j} - s_{i,j'}) + \lambda\|\Theta\|, \tag{3}$$

where $\sigma(\cdot)$ is the sigmoid function, $j'$ is the negative sample, $\Theta$ denotes all trainable parameters for the graph auto-encoder, and $\lambda$ is the coefficient for the L2 regularization.

*3.2.2 Graph Pruning.* After training the graph auto-encoder, we obtain an embedding for each item, which preserves its relational information with other items. We use these embeddings to prune the graph, to achieve a more concise graph with cleaner relations. For each item $i$, we rank all its first-order neighbors on the graph according to the pair-wise inner-product scores. We deem that the pairs with higher scores are more likely to be reliable and of high quality, while the pairs with lower scores are prone to be noisy or redundant. For each item, we remove $\beta\%$ of its first-order relations that have the lowest scores, where $\beta$ is a hyper-parameter to control the pruning ratio.

After the graph pruning, we obtain a smaller graph, denoted as $\tilde{\mathcal{G}}$. We deem that our relation pruning method is a self-supervised bootstrapping approach to denoising the graph data. More importantly, the pruning is dependent on the item embeddings that are learned from high-order graph propagation, justifying that the remained item-item relations are more reliable.

## 3.3 Pre-train Framework

The architecture of the pre-train framework encompasses two parts: multimodal encoder and pre-train objective.

*3.3.1 Multimodal Encoder.* We inherit the backbone of BLIP [22] to encode the image and text input. Specifically, we use the visual transformer [8] as the image encoder, which inputs a sequence of image patches and uses the $[CLS]$ token to represent the feature of the whole. In parallel, we use BERT [5] as the text encoder, which inputs the sequence of textual tokens and makes use of the $[CLS]$

token to represent the entire text. Given the image $v_i$ and text $t_i$ of an item $i$, the multimodal encoder outputs its visual and textual representations, denoted as $\mathbf{v}_i$ and $\mathbf{t}_i$, respectively.

*3.3.2 Pre-train Objective.* In order to make the pre-train model capture both individual items' semantics and cross-item relational information, we jointly optimized two pre-train losses, *i.e.,* image-text contrastive loss (ITC) and cross-item contrastive loss (CIC).

- **ITC Loss.** We inherit it from BLIP [22] and ALBEF [23], targeting at the cross-modality alignment between image and text. The ITC loss for item $i$ is represented as:

$$\mathcal{L}_i^{itc} = \text{Contrast}(\mathbf{v}_i, \mathbf{t}_i), \tag{4}$$

where $\mathbf{v}_i, \mathbf{t}_i$ are the visual and textual representations output from the multimodal encoder. Contrast$(\cdot)$ denotes the contrastive loss function implemented by ALBEF [23], where a momentum encoder and soft labels from the momentum encoders are utilized. The details of the equations for contrastive loss are provided in the supplementary part.

- **CIC Loss.** We employ the CIC loss to model the cross-item relations by pulling close a pair of related items in the representation space. For every pair of items $(i, j) \in \tilde{\mathcal{G}}$, we obtain their image and text representations $(\mathbf{v}_i, \mathbf{t}_i, \mathbf{v}_j, \mathbf{t}_j)$ through the multimodal encoder. We then harness a contrastive loss to form the CIC, denoted as:

$$\mathcal{L}_{i,j}^{cic} = \text{Contrast}(\mathbf{v}_i, \mathbf{t}_j) + \text{Contrast}(\mathbf{t}_i, \mathbf{v}_j), \tag{5}$$

where Contrast$(\cdot)$ is the contrastive loss same with Equation 4. The difference between CIC and ITC is that the image-text contrastive pair of CIC is from different items, while the contrastive pair of ITC is from the same item.

- **Optimization.** We first finetune BLIP on the image-text pairs of our pre-training dataset, to fully utilize the multiple training objectives of the original BLIP (*i.e.,* ITM and LM) and capture the domain characteristics. Thereafter, we take the finetuned multimodal encoder to initialize our CIRP encoder. The overall pre-train objective of CIRP is obtained by combining the ITC and CIC loss, denoted as:

$$\mathcal{L} = \mathbb{E}_{(i,j)\sim\tilde{\mathcal{G}}} \mathcal{L}_i^{itc} + \mathcal{L}_j^{itc} + \mathcal{L}_{i,j}^{cic}. \tag{6}$$

It is worth noting that even though the finetuned BLIP already well captures the semantics of image and text, the ITC loss is still essential during our pre-training. Empirical study (please refer to Section 4.3) shows that only using the CIC loss could mislead the pre-training and destroy the learned semantic modeling capability.

## 3.4 Product Bundling

The main objective of our pre-training framework is to perform the task of multimodal product bundling. Since this work focuses on learning high-quality item representations through the pre-train model, we prefer a representative but simple model for product bundling, thus preventing any potential bias introduced by the downstream model. Hence, we follow the previous work [40] and employ ItemKNN [36].

Specifically, given a partial bundle $P = \{i_n\}_{n=1}^{N_p}$ that consists of a set of $N_p$ seed items, the task of product bundling aims to complete

Table 1: The statistics of the three e-commerce datasets for both pre-train and downstream task of product bundling.

| Datasets | Pre-train | | Downstream Task | | |
|---|---|---|---|---|---|
| | #items | #relations | #bundles | #items | #avg. size |
| Clothing | 236,387 | 278,259 | 1,910 | 4,487 | 3.31 |
| Electronic | 178,443 | 658,446 | 1,750 | 3,499 | 3.52 |
| Food | 54,323 | 86,373 | 1,784 | 3,767 | 3.58 |
| Total | 469,153 | 1,023,078 | 5,444 | 11,753 | 3.47 |

the bundle by selecting the most probable items from a candidate set of items. As shown in Figure 2, we first use the pre-trained model extract the visual and textual representations of a given item, denoted as:

$$\mathbf{v}_i, \mathbf{t}_i = \Phi(v_i, t_i; \Theta). \quad (7)$$

Next, we obtain the multimodal representation of each item by averaging both visual and textual features $\mathbf{x}_i$, formally written as:

$$\mathbf{x}_i = \frac{1}{2}(\mathbf{v}_i + \mathbf{t}_i). \quad (8)$$

As following, we aggregate the representations of items in the partial bundle using the simple average pooling strategy, resulting in a partial bundle representation $\mathbf{p}$, represented as:

$$\mathbf{p} = \frac{1}{N_P} \sum_{i \in P} \mathbf{x}_i. \quad (9)$$

Then we rank all the candidate items $\mathcal{I}$ based on the affinity score between $\mathbf{p}$ and $\mathbf{x}_i$, where $i \in \mathcal{I}$. Finally, we get the top-k items $\hat{\mathcal{I}}$ that are most probably be selected to complete the input partial bundle, denoted as:

$$\hat{I} = \underset{\hat{I} \subset \mathcal{I}; |\hat{I}| = k}{\arg\max} \sum_{i \in \hat{I}} \cos(\mathbf{x}_i, \mathbf{p}), \quad (10)$$

where $\cos(\cdot, \cdot)$ denotes the cosine similarity function, and $k$ is the hyper-parameter during testing. It should be noted that ItemKNN is a model-free method, which means it includes no auxiliary trainable parameters while only relying on the item representations extracted by the pre-train model. We acknowledge there are more model-based bundle recommendation models, such as auto-encoder models [31]. However, given the extremely small scales of the datasets [39, 40], the bundle construction models would severely overfit to the small datasets. We leave the study of using model-based product bundling methods for future work.

## 4 EXPERIMENTS

We conduct extensive experiments on e-commerce datasets to answer the following research questions:

- How does our proposed CIRP model perform on the task of product bundling, compared to various baseline methods?
- Whether are the key modules in CIRP effective?
- What are the key and interesting properties of CIRP?

## 4.1 Experimental Settings

*4.1.1 Datasets and Evaluation Metrics.* Even though product bundling is mature in business, there are few public datasets that are suitable for product bundling, especially in the multimodal setting. We

utilize the Amazon review dataset for pre-training and the three bundle datasets, *i.e.,* Clothing, Electronic, and Food [39, 40], for product bundling. The statistics of the three datasets are shown in Table 1. We unify all three datasets into a big one for pre-training. For the downstream task of product bundling, we use the same pre-trained model and evaluate it on the three datasets separately. Since ItemKNN does not need training, we use the whole bundle dataset for testing. We follow the previous work [39, 40] and employ the Recall@k and NDCG@k to evaluate the performance of product bundling, where $k \in \{10, 20\}$.

*4.1.2 Baselines.* Since we are pioneering in using pre-train models for product bundling, there are few baselines that exactly match our task on these datasets. Therefore, we consider three streams of item representation learning baselines according to the input information, *i.e.,* relation-only (REL-only), semantics-only (SEM-only), and relation and semantics (REL-SEM). We briefly introduce these baselines, and their implementation details are presented in the supplementary notes.

- **REL-only** methods only use the relational data, without any image or text features. They can be broadly categorized into two types: *non-sequential* (MFBPR, LightGCN, and SGL) and *sequential* methods (Caser, GRU4Rec, and SASRec). MFBPR [35] is the pre-train model used by Sun *et al.* [40]. We additionally adopt two more powerful non-sequential methods of LightGCN [13] and SGL [47]. For sequential methods, we implement three classical models: the CNN-based method Caser [41], the RNN-based method GRU4Rec [15], and the self-attention-based method SASRec [18]. To be noted, given the sequential essence, during the product bundling, we treat the input partial bundle as a sequence and directly use the pre-trained model to get the partial bundle representation $\mathbf{p}$, which is then used to rank all the items. This setting is designed to faithfully and maximally retain the sequential methods' capabilities.
- **SEM-only** models only leverage the item semantic information (*i.e.,* the image and text), without any item-item relational data. We implement the leading methods CLIP [33] and BLIP [22], of which the parameters are fixed. We also finetune CLIP and BLIP on our image-text pairs, resulting in the CLIP-FT and BLIP-FT methods.
- **REL-SEM** approaches utilize both relational and semantic data, including feature fusion (FF-) methods and graph learning (GL-) methods. Both types of methods require pre-extracted multimodal features, where we use the best-performing BLIP-FT as the feature extractor. For feature fusion methods, we concatenate the multimodal features with relational features. We have FF-LightGCN and FF-SGL, using the relational embedding obtained by the REL-only model LightGCN and SGL, respectively. For graph learning methods, we use the multimodal features to initialize the node embedding and employ GCN [20] and GCL (Graph Contrastive Learning [47]) as the graph model, resulting in the GL-GCN and GL-GCL baselines.

*4.1.3 Implementation Details.* Our model is initialized from the pre-trained BLIP [22] with BERT as text encoder and ViT-Base [8] as image encoder. The BLIP is finetuned on our image-text pairs for 10 epochs and our pre-train model is trained on the relation

**Table 2: The overall performance comparison, where R is short for Recall and N for NDCG. The strongest baselines are underlined, and "%Improv." indicates the relative improvement compared to the strongest baselines.**

| Models | | Clothing | | | | Electronic | | | | Food | | | |
|---|---|---|---|---|---|---|---|---|---|---|---|---|---|
| | | R@10 | N@10 | R@20 | N@20 | R@10 | N@10 | R@20 | N@20 | R@10 | N@10 | R@20 | N@20 |
| REL-only | MFBPR | 0.0421 | 0.0239 | 0.0664 | 0.0300 | 0.0914 | 0.0494 | 0.1216 | 0.0571 | 0.0906 | 0.0492 | 0.1331 | 0.0599 |
| | LightGCN | 0.0676 | 0.0377 | 0.1021 | 0.0465 | 0.1001 | 0.0584 | 0.1405 | 0.0686 | 0.1321 | 0.0711 | 0.1802 | 0.0833 |
| | SGL | 0.0890 | 0.0504 | 0.1234 | 0.0591 | 0.1008 | 0.0588 | 0.1443 | 0.0698 | 0.1446 | 0.0773 | 0.1878 | 0.0881 |
| | Caser | 0.0092 | 0.0044 | 0.0139 | 0.0056 | 0.0289 | 0.0152 | 0.0428 | 0.0187 | 0.0215 | 0.0119 | 0.0307 | 0.0141 |
| | GRU4Rec | 0.0091 | 0.0043 | 0.0159 | 0.0060 | 0.0283 | 0.0153 | 0.0407 | 0.0184 | 0.0222 | 0.0120 | 0.0356 | 0.0154 |
| | SASRec | 0.0135 | 0.0072 | 0.021 | 0.0091 | 0.0337 | 0.0179 | 0.0509 | 0.0222 | 0.0171 | 0.0083 | 0.0280 | 0.0110 |
| SEM-only | CLIP | 0.2926 | 0.1942 | 0.3704 | 0.2138 | 0.0898 | 0.0536 | 0.1363 | 0.0653 | 0.2867 | 0.1995 | 0.3524 | 0.2159 |
| | CLIP-FT | 0.2982 | 0.1928 | 0.3799 | 0.2134 | 0.1006 | 0.0628 | 0.1448 | 0.0740 | 0.2633 | 0.1842 | 0.3152 | 0.1971 |
| | BLIP | 0.3176 | 0.2101 | 0.4041 | 0.2319 | 0.1032 | 0.0631 | 0.1518 | 0.0754 | 0.2956 | 0.2038 | 0.3444 | 0.2161 |
| | BLIP-FT | 0.3483 | 0.2347 | 0.4303 | 0.2551 | 0.1271 | 0.0801 | 0.1739 | 0.0919 | 0.3164 | 0.2189 | 0.3758 | 0.2339 |
| REL-SEM | FF-LightGCN | 0.2699 | 0.1809 | 0.3246 | 0.1947 | 0.1457 | 0.0872 | 0.1982 | 0.1005 | 0.2607 | 0.1766 | 0.3128 | 0.1899 |
| | FF-SGL | 0.2900 | 0.1924 | 0.3458 | 0.2066 | 0.1613 | 0.0949 | 0.2160 | 0.1087 | 0.2727 | 0.1812 | 0.3273 | 0.1951 |
| | GL-GCN | 0.3145 | 0.2149 | 0.3853 | 0.2329 | 0.1571 | 0.0947 | 0.2075 | 0.1073 | 0.2822 | 0.1953 | 0.3403 | 0.2100 |
| | GL-GCL | 0.3131 | 0.2082 | 0.3790 | 0.2250 | 0.1624 | 0.0949 | 0.2195 | 0.1093 | 0.2778 | 0.1877 | 0.3400 | 0.2035 |
| Ours | CIRP | **0.3906** | **0.2651** | **0.4866** | **0.2893** | **0.2103** | **0.1210** | **0.2801** | **0.1385** | **0.3430** | **0.2304** | **0.4140** | **0.2481** |
| | %Improv. | 12.14 | 12.95 | 13.08 | 13.41 | 29.50 | 27.50 | 27.61 | 26.72 | 8.41 | 5.25 | 10.16 | 6.07 |

pairs for 10 epochs using 4*A5000 (24G) GPUs. We use AdamW optimizer [28] with a weight decay of 0.05. The learning rate is set to $3.0 \times 10^{-5}$ and decayed linearly with a ratio of 0.9 at the end of every training epoch. The batch size is 24 for BLIP finetuning and 16 for CIRP. For relation pruning, we train the graph auto-encoder and perform graph pruning on each dataset separately. For each dataset, we split the pre-train item relations into training, validation, and testing sets with the ratio of 8:1:1. We search for the best hyper-parameter of LightGCN with grid search. Wherein, learning rate is searched from $1.0 \times 10^{-3}$ to $1.0 \times 10^{-4}$, and weight decay is searched from $1.0 \times 10^{-4}$ to $1.0 \times 10^{-7}$. The embedding size is tuned in the range of {16, 32, 64, 128 }, and the number of propagation layers is searched from {1, 2, 3}. After determining the best hyper-parameter for each dataset, we train the graph auto-encoder with full data using the best hyper-parameter, and then we use it for graph pruning. The implementation details of all the baseline methods, including REL-only, SEM-only, and REL-SEM baselines, are presented in the supplementary file.

## 4.2 Performance Comparison (RQ1)

Table 2 explicates the overall performance of our method CIRP and the baseline methods. We have the following observations. First, our approach CIRP beats all the baselines, especially in the Electronic data, which achieves over 25% relative performance improvements. Second, SEM-only methods are competitive and outperform all the REL-only methods, demonstrating their strong generalization capability. Interestingly, for non-sequential REL-only or SEM-only baselines, stronger models yield better performance. This justifies that either relational or semantic data pertains essential information for product bundling, and the performance is positively correlated with

**Table 3: The ablation study of CIRP. ITC and CIC are the pre-train objectives, and RP represents *Relation Pruning*.**

| Models | Clothing | | Electronic | | Food | |
|---|---|---|---|---|---|---|
| | R@20 | N@20 | R@20 | N@20 | R@20 | N@20 |
| BLIP | 0.4041 | 0.2319 | 0.1518 | 0.0754 | 0.3444 | 0.2161 |
| -ITC&CIC | 0.4303 | 0.2551 | 0.1739 | 0.0919 | 0.3758 | 0.2339 |
| -ITC | 0.4162 | 0.2264 | 0.2224 | 0.1020 | 0.3309 | 0.1860 |
| -RP | 0.4738 | 0.2801 | 0.2771 | 0.1372 | 0.4030 | 0.2392 |
| CIRP | 0.4866 | 0.2893 | 0.2801 | 0.1385 | 0.4140 | 0.2481 |

the pattern modeling capability on the pre-training dataset. However, for sequential-based REL-only methods, their performances are significantly worse than their non-sequential counterparts. This may because: 1) the items within a bundle are insensitive to the sequential order; and more importantly 2) the sequence in the pre-train dataset is too short. Third, even though our method surpasses REL-only and SEM-only methods, other REL-SEM models struggle to do it. This observation further strengthens that the pre-train framework plays a pivotal role in maximizing the utility of all the data. Finally, the effects of each type of data vary across different datasets. For example, when comparing REL-SEM and SEM-only methods, after incorporating the relational data, the performance gain on *Electronic* dataset is much more than those on *Clothing* and *Food* datasets. This indicates that relations are more prominent in *Electronic* while semantics are more crucial for *Clothing* and *Food* when bundling products.

**Table 4: Model performance under the cold-start setting.**

| Models | setting | Clothing | | Electronic | | Food | |
|---|---|---|---|---|---|---|---|
| | | R@20 | N@20 | R@20 | N@20 | R@20 | N@20 |
| BLIP | - | 0.4041 | 0.2319 | 0.1518 | 0.0754 | 0.3444 | 0.2161 |
| BLIP-FT | warm | 0.4303 | 0.2551 | 0.1739 | 0.0919 | 0.3758 | 0.2339 |
| | cold | 0.4236 | 0.2544 | 0.1736 | 0.0961 | 0.3768 | 0.2302 |
| CIRP-RP | warm | 0.4738 | 0.2801 | 0.2771 | 0.1372 | 0.4030 | 0.2392 |
| | cold | 0.4671 | 0.2845 | 0.2462 | 0.1243 | 0.4006 | 0.2434 |

## 4.3 Ablation Study (RQ2)

We progressively remove modules of CIRP and curate three ablated models to demonstrate the effectiveness of each key module; where "-ITC&CIC" represents removing both losses of ITC and CIC, which is identical with the BLIP-FT baseline. "-ITC" means only removing the ITC loss while keeping the CIC loss. "-RP" corresponds to the model variant without any relation pruning. The results of the ablated models are presented in Table 3. First, "-ITC&CIC" is worse than CIRP, illustrating that the loss combination of ITC and CIC is reasonable and effective. Second and interestingly, "-ITC" underperforms not only CIRP but also "-ITC&CIC". This shows that the cross-modal alignment loss (ITC) plays a crucial role in cross-item relational modeling. Only applying CIC loss cannot well capture the cross-item relations, even worse, it will corrupt the semantic modeling capability pertained within the backbone model. Jointly optimizing both ITC and CIC loss is key to the success of CIRP. Third, "-RP" is slightly worse than CIRP, showing that relation pruning is able to enhance the performance.

## 4.4 Model Study (RQ3)

We are interested in multiple key properties of CIRP and conduct multiple model studies to investigate the details.

*4.4.1 Effects on Cold-start Items.* To evaluate our model's generalization capability on cold-start items, we remove all the items, which exist in both pre-train and downstream datasets, from the pre-train dataset. Afterwards, we re-train our model and baselines, and evaluate them on the product bundling task. To be noted, RELonly and SEM-REL models cannot deal with cold-start items since item relations are necessary for them. Given that, we only implement the strongest SEM-only baseline BLIP-FT. The results in Table 4 show that cold-start has little impact on both BLIP-FT and CIRP-RP (with no relation pruning). We can derive that multimodal pre-train models' generalization capability, which has been verified in various vision-language downstream tasks, does also apply to the cross-item relational task of product bundling. Moreover, the performance gap between the baseline model and our method is consistent no matter whether in warm or cold-start setting. This implies that the cross-item relational patterns have approximately the same level of generalization capability with semantic patterns.

*4.4.2 Effects w.r.t. Varying Relation Pruning Ratio.* In order to quantitatively illustrate the denoise and acceleration effects of relation pruning, we progressively increase the relation pruning ratio from 10% to 90% and record the corresponding performance and training

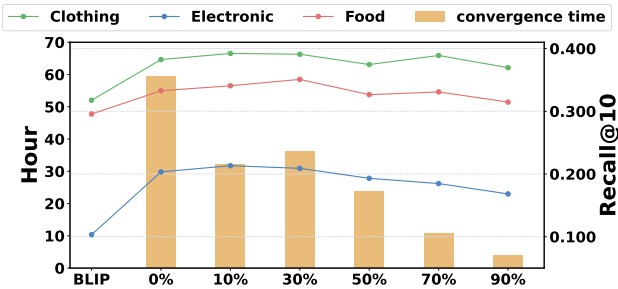

**Figure 3: Analysis of how varying relation pruning rate affect the pre-train efficiency and product bundling performance.**

**Table 5: Representation distribution comparison between the random items ($\bar{s}_{avg}$) and intra-bundle items ($\bar{s}_{intra}$).**

| Models | Clothing | | Electronic | | Food | |
|---|---|---|---|---|---|---|
| | $\bar{s}_{avg}$ | $\bar{s}_{intra}$ | $\bar{s}_{avg}$ | $\bar{s}_{intra}$ | $\bar{s}_{avg}$ | $\bar{s}_{intra}$ |
| BLIP | 0.4718 | 0.6143 | 0.5334 | 0.5992 | 0.5375 | 0.6212 |
| BLIP-FT | 0.2848 | 0.4484 | 0.3054 | 0.3683 | 0.3454 | 0.4605 |
| CIRP | 0.2843 | 0.5256 | 0.2732 | 0.4069 | 0.4568 | 0.6173 |

time (on 4*A5000 GPUs) under each setting, as shown in Figure 3. First, with the pruning ratio increasing, the performance of our model first goes up and then drops. This phenomenon aligns well with our hypothesis that pruning prioritizes noisy relations. When the pruning ratio keeps increasing, more benign relations start to be removed, as a result, the performance curves turn down. Second, with the pruning ratio increasing, the convergence time drops significantly. For example, when the pruning ratio is 10%, the convergence time is only half of the original cost, while the performance slightly increases. When the pruning ratio is 90%, the convergence time decreases to 10% of the original setting, with only a marginal performance drop. The results show that the relation pruning can significantly improve the pre-train efficiency with marginal or even no performance drop.

*4.4.3 Representation Learning Analysis.* In order to demystify the working mechanism of our method, we quantitatively study the representation characteristics. Specifically, we make statistics of the average cosine similarity of random item pairs and the item pairs within the same bundle, denoted as $\bar{s}_{avg}$ and $\bar{s}_{intra}$, respectively. We compare three models of BLIP, BLIP-FT, and CIRP, the results of which are depicted in Table 5. Generally, the intra-bundle similarity score is higher than that of random item pairs, for all sthree methods and datasets. This implies that the items within the same bundle are semantically close to each other. When comparing BLIP and BLIP-FT, the average item similarity score decreases. This shows that finetuning BLIP enforces the items scatter and occupy as much space as possible, thus endowing the model's improved representation capability to discriminate trivial differences, which could not be captured before finetuning. For the intra-bundle cross-item similarity, it decreases significantly on BLIP-FT while increases on

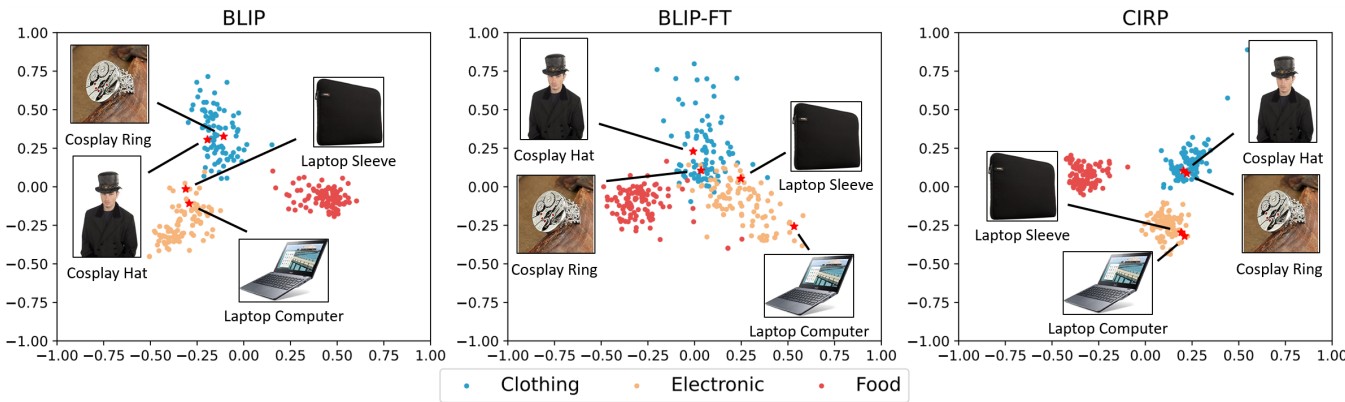

**Figure 4: Qualitative visualization of how the item representation space shift after applying the semantic pre-training (BLIP-FT) and cross-item relational pre-train (CIRP).**

**Table 6: Product bundling cases: given a query item, we show the ground-truth item's ranking made by various methods.**

| Query Item | Ground-truth | Ranking | | |
|---|---|---|---|---|
| | | BLIP | BLIP-FT | CIRP |
| Basketball Shoes | Running Shoes | 87 | 7 | 5 |
| Streaming Media Players | HDMI Cable | 856 | 213 | 11 |

CIRP conversely, explicitly illustrating that our proposed cross-item relational pre-train can re-gather the items within the same bundle.

To further qualitatively demonstrate how the embedding space differs *w.r.t.* different models, we randomly select 20 bundles and project their included items' embeddings into a 2D plane, as shown in Figure 4. Clearly, we can find out that BLIP-FT makes the gathered items expand to a larger space, however, it simultaneously pushes away the items within the same bundle. By applying cross-item relational pre-train, the item distribution becomes more compact, especially for items within the same bundle. This observation aligns well with the quantitative results presented in Table 5.

*4.4.4 Case Study.* To directly demonstrate how the model performs on product bundling, we pick two pairs of items and present them in Table 6. Each item pair is from the same bundle, where the first item is provided as the query while the second item is the ground-truth item for product bundling. The number in Table 6 corresponds to the ranking of the second item in the candidate set for different

models. We can see that by finetuning BLIP and applying cross-item relational pre-training, the ranking of the second item keeps increasing. Interestingly, the two cases illustrate two representative bundling strategies. The first item pair is curated based on semantic similarity, that's why BLIP-FT already works very well. The second item pair is bundled based on implicit relation, *i.e.,* functionally compatible, which is hard to be identified from the semantic features. That's why both BLIP and BLIP-FT fail to recognize the relations between these two items. However, our method CIRP is able to capture such implicit relations and ranks the *HDMI Cable* high when given a *Streaming Media Player* as the query.

## 5 CONCLUSION AND FUTURE WORK

We explored an interesting task of integrating the cross-item relations into a multimodal pre-train model for product bundling. We developed a novel framework CIRP that leverages both intra-item cross-modality contrastive loss (ITC) and cross-item contrastive loss (CIC). More importantly, to alleviate the noise in the relations as well as reduce the training cost, we designed a relation pruning method to enhance and accelerate the pre-train. We tested our method on large-scale e-commerce datasets and evaluated its performance on three product bundling datasets. Experimental results demonstrated the effectiveness and efficiency of our method.

Despite the great progress, there are multiple potential directions to be explored in the future. First, current work only focuses on the first-order direct relations among items, while the higher-order relations that contain additional crucial information should also be considered. Second, our method only considers one type of item-item relations, and more types of heterogeneous relations could be more helpful yet challenging to model. Therefore, integrating heterogeneous relations, such multiple behaviors, cross-domain relations, and relations from knowledge graph, into the pre-train model should be investigated in the future. Third, cross-item relational pre-training is a general paradigm and can be generalized to more downstream tasks, such as session-based and sequential recommendations. Finally, it is worthwhile to embrace the powerful LLMs, for example, we can directly use LLMs for product bundling or incorporate relations into the LLMs to boost the performance.

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
