# OpenReview forum: "CIRP: Cross-Item Relational Pre-training for Multimodal Product Bundling"
_acmmm.org/ACMMM/2024/Conference — MM2024 Poster_

### Official Review · Reviewer_4Z8R · 2024-05-22

**Rating:** 5
**Confidence:** 3

**Summary:**

This paper introduces a novel framework CRIP aimed at improving item representation learning for product bundling in e-commerce.

**Strengths:**

1. The paper addresses multimodal product bundling, an intriguing and relevant topic.

2. This innovative approach fills a significant gap in the current methodologies which focus primarily on semantic modeling without adequately addressing cross-item relations.

3. CIRP employs both cross-item contrastive loss (CIC) and individual item's image-text contrastive loss (ITC) as pre-training objectives. This dual approach ensures that the model captures both the semantic details of individual items and the relational aspects between items, making it highly effective for product bundling tasks.

3. It demonstrates significant improvements over baseline approaches and provides a detailed analysis.

4. The proposed relation graph pruning approach significantly speeds up pre-training time.

5. The paper is well-written and well-organized

6. The authors have committed to releasing the code.

**Limitations:**

Although the authors employ graph pruning to reduce training costs significantly, fully fine-tuning the multimodal encoders remains costly. The authors might consider exploring more efficient methods for this process in future research, such as [1].

[1] "IISAN: Efficiently Adapting Multimodal Representation for Sequential Recommendation with Decoupled PEFT." arXiv preprint arXiv:2404.02059 (2024).

**Suitability:**

3

---

### Official Review · Reviewer_vev3 · 2024-05-24

**Rating:** 4
**Confidence:** 2

**Summary:**

This paper proposes a novel CIRP method for item representation learning in product bundling. This method leverages both the cross-item contrastive loss (CIC) and individual item’s image-text contrastive loss (ITC) as the pre-train objectives. Extensive experiments on three publicly datasets demonstrate the effectiveness of the proposed method.

**Strengths:**

1. The motivation is novel. The paper achieves a simple pre-training framework to naturally integrate
cross-item relations into the multimodal encoder by the CIC loss, while preserving the in-depth aligned multimodal semantics by the ITC loss.
2. The experiment results demonstrate the effectiveness of the proposed method, especially for cold-start items.
3. The manuscript is easy to follow as it is well-organized.

**Limitations:**

1. A negative sample in ITC loss may be a positive sample in CIC loss. So, the design of negative samples is very important. Further clarification is needed.
2. The pre-training time seems to have a linear relationship with the number of datasets, and the pre-trained model seems unsuitable for downstream new datasets.
3. Is relation pruning pre-trained separately for each dataset?
4. In table 1, how is the bundle of each dataset constructed?
5. How can different datasets be mutually enhanced through pre-training?

**Suitability:**

3

---

### Official Review · Reviewer_r9R3 · 2024-05-25

**Rating:** 4
**Confidence:** 3

**Summary:**

The paper presents a novel framework for enhancing product bundling in e-commerce through improved item representation learning, aiming to develop high-quality item representations capturing both individual semantics and cross-item relations for effective product bundling as Current methods (feature fusion and graph learning) struggle with cross-modal alignment and cold-start items. They Introducing Cross-Item Relational Pre-training (CIRP), which uses a multimodal encoder and contrastive losses to integrate relational information into item representations.

**Strengths:**

1.Improved Cross-Modal Alignment:
Unlike previous methods that separately learn features from multiple modalities, CIRP ensures in-depth cross-modal alignment and enhancement by using multimodal encoders and contrastive losses .

2.Effective Handling of Cold-Start Items:
CIRP can generate relation-aware representations even for new items that do not have pre-existing relational data, addressing a significant limitation of earlier methods .

3.Noise Reduction and Efficiency:
The framework includes a relation pruning module that removes noisy and redundant relations, which not only reduces computational cost but also improves the efficiency of the pre-training process. This allows CIRP to maintain high performance while significantly cutting down on pre-training time .

4.Superior Performance in Product Bundling:
Experimental results on three large-scale e-commerce datasets demonstrate that CIRP outperforms leading representation learning methods, proving its effectiveness in real-world applications .

5.Scalability:
By efficiently handling a large amount of relational data and leveraging advanced multimodal encoders, CIRP is well-suited for the ever-growing number of products in e-commerce platforms, making it highly scalable .

**Limitations:**

1. Lack of Novelty: The proposed pre-training item representation lacks innovation, as similar ideas have been explored in previous works [1,2]. The authors merely apply two existing models without modification for separate modality representation and align them using contrastive loss.

   [1] Hou, Yupeng, et al. "Towards universal sequence representation learning for recommender systems." Proceedings of the 28th ACM SIGKDD Conference on Knowledge Discovery and Data Mining. 2022.

   [2] Li, Youhua, et al. "Multi-Modality is All You Need for Transferable Recommender Systems." arXiv preprint arXiv:2312.09602 (2023).

2. Figure 2 lacks a clear description of the icons representing the inputs to the separate multimodal encoder. Additionally, the absence of a 'cls' token in the image encoder input, as mentioned in section 3.3.1, is not addressed. Recall@10 is used to evaluate the effect of the relation pruning rate, but Recall@20 is used in most other tables, leading to inconsistency.


3. Cold-Start Item Experiment: In section 4.4, the comparison for cold-start items is only made against the strongest SEM-only baseline BLIP-FT, which may not be the most appropriate choice for cold-start scenarios. A more comprehensive comparison with other baselines would provide better insights.

**Suitability:**

2

---

### Official Review · Reviewer_XTxG · 2024-05-26

**Rating:** 3
**Confidence:** 3

**Summary:**

this paper introduces a new framework designed to improve item representation learning in the context of product bundling. The approach leverages multimodal encoders and novel pre-training objectives to capture both individual item semantics and cross-item relationships, particularly focusing on the cold-start issue where items lack explicit relational data. The key components of CIRP include cross-item contrastive loss (CIC) and individual item's image-text contrastive loss (ITC), along with a relation pruning module to filter out noisy or redundant relations. Experimental results show that CIRP outperforms several baselines, demonstrating its effectiveness in enhancing the generalization capability of models on cold-start items.

**Strengths:**

Strengths of the CIRP model include its ability to integrate cross-item relational modeling into multimodal encoders, its performance improvements in the cold-start setting, and the successful demonstration of the importance of both ITC and CIC losses in improving item representations. The model also highlights the significance of relation pruning in refining the learned representations.

**Limitations:**

1 Figure 1 does not contain any motivation about why the authors chose this style for learning representations. Also I think in this important figure, the authors should give some insights about the Multimodal Product Bundling, where the proposed framework could apply.
2 Vision Language Model is a keyword, but its connection is weak to the context of this paper.
3 The technical contribution is too weak for MM. In my opinion, the authors only propose combining two representation learning in a pretrain model for recommendation. Also, the pre-training work could be replaced by LLM models easily.

**Suitability:**

2

---

### Meta-Review · Program_Chairs · 2024-07-13

**Recommendation:** Accept (Poster)
**Confidence:** 4

**Metareview:**

This paper proposes a novel framework for item representation in product bundling, Cross-Item Relational Pre-training (CIRP). The topic is relevant and important, well suited to the conference. Even though one reviewer is not convinced, the other three align towards acceptance, due to its performance and the scalability of the approach.